# The Role of Docosahexaenoic Acid (DHA) on Cognitive Functions in Psychiatric Disorders

**DOI:** 10.3390/nu11040769

**Published:** 2019-04-02

**Authors:** Valentina Ciappolino, Alessandra Mazzocchi, Andrea Botturi, Stefano Turolo, Giuseppe Delvecchio, Carlo Agostoni, Paolo Brambilla

**Affiliations:** 1Fondazione IRCCS Ca’ Granda-Ospedale Maggiore Policlinico, Department of Neurosciences and Mental Health, 20122 Milan, Italy; valentina.ciappolino@policlinico.mi.it (V.C.); paolo.brambilla1@unimi.it (P.B.); 2Department of Clinical Sciences and Community Health, University of Milan, 20122 Milan, Italy; alessandra.mazzocchi@unimi.it; 3Neurologic Clinic, Fondazione IRCCS Istituto neurologico Carlo Besta, 20122 Milan, Italy; Andrea.Botturi@istituto-besta.it; 4Fondazione IRCCS Ca’ Granda-Ospedale Maggiore Policlinico, Pediatric Nephrology, Dialysis and Transplant Unit, 20122 Milan, Italy; stefano.turolo@policlinico.mi.it; 5Department of Pathophysiology and Transplantation, University of Milan, 20122 Milan, Italy; giuseppe.delvecchio@unimi.it; 6Fondazione IRCCS Ca’ Granda-Ospedale Maggiore Policlinico, Pediatric Intermediate Care Unit, 20122 Milan, Italy

**Keywords:** *n*-3 long chain polyunsaturated fatty acids, docosahexaenoic acid, psychiatric disorders, cognition, cognitive symptoms

## Abstract

Cognitive impairment is strongly associated with functional outcomes in psychiatric patients. Involvement of *n*-3 long chain polyunsaturated fatty acid (*n*-3 LC-PUFA), in particular docosahexaenoic acid (DHA), in brain functions is largely documented. DHA is incorporated into membrane phospholipids as structural component, especially in the central nervous system where it also has important functional effects. The aim of this review is to investigate the relationship between DHA and cognitive function in relation to mental disorders. Results from few randomized controlled trials (RCTs) on the effects of DHA (alone or in combination) in psychotic, mood and neurodevelopmental disorders, respectively, suggest that no conclusive remarks can be drawn.

## 1. Introduction 

The growing number of studies on the association of *n*-3 long chain polyunsaturated fatty acids (*n*-3 LC-PUFAs) on cognitive function and mental health, especially psychiatric illness, published in the last decades reflects growing interest in this area of research. Given the global market for *n*-3 PUFA products, it is of public importance that there is a more conclusive picture as to whether their supplementation improves cognitive performance. Therefore, we conducted a review of randomized placebo-controlled trials (RCTs) which examined the role of *n*-3 PUFAs, in particular docosahexaenoic acid (DHA), in maintaining or in treating cognitive functions in different mental disorders. Cognitive functioning can be defined as a collection of different abilities including memory, language, attention, perception, problem solving and mental imagery. These cognitive and practical competencies are the key to enhancing individual and community well-being and to living a life of purpose [1]. Cognition can be sub-divided into different domains. According to the Diagnostic and Statistical Manual of Mental Disorders (DSM-5) [2] there are six key domains: complex attention, executive function, learning and memory, language, perceptual-motor, and social cognition. Each cognitive domain is divided into several more specific cognitive subtypes (Figure 1). Attention can be considered a fundamental cognitive domain because it permits one to accept, recall and maintain different and complex information. More specifically, complex attention comprises several abilities, including the capacity to sustain attention, the capacity to selectively maintain attention to a specific stimulus, the capacity to shift attention, and the capacity to put attention to various stimuli at the same time [2,3]. Similarly, executive functions include as sub domains planning, decision making, working memory, responding to feedback, inhibition, and flexibility [2]. Moreover, learning and memory represent mental functions that allow people to acquire, retain, and recall new impressions, information and sensations after they have been experienced. Memory can be divided into three stages: sensory, short-term and long-term. Information processing starts with sensory memory, moves to short-term memory and eventually transfers to long-term memory. Then, the function of long-term memory is to store data through various categorical models or systems [4]. Interestingly, within the cognitive domains there is also language, which can be defined as the capacity of naming, word searching, grammar using, and comprehension; more specifically two different aspects of language are identified as expressive and receptive [2]. Another cognitive domain is the perceptual-motor function comprising visual perception, visuospatial-construction, perceptual-motor coordination, and performance [2]. The dysfunction of the perceptual-motor domain, occurring for example with neurologic injury, can result in apraxias, agnosia, hemineglect, and Balint syndrome [5]. Finally, the last cognitive domain is social cognition, which is the ability to recognize emotions in social situations, to have a theory of mind, to feel empathy, to make an attributional style, and to have an insight into something. It is involved in social and vocational functioning and interpersonal relationships [6].

### 1.1. Cognitive Impairment in Psychiatric Disorders

Cognitive impairments are reported as one of the main features of psychiatric disorders and they have a major negative impact on psychosocial functioning of patients [7,8]. Indeed, several mental disorders, such as major depressive disorder (MDD), bipolar disorder (BD), schizophrenia (SZ), attention deficit/hyperactivity disorder (ADHD), post-traumatic stress disorder (PTSD), and obsessive-compulsive disorder (OCD) were consistently found to have impairments in cognitive functions involving verbal memory, executive functions, attention and processing speed, with different levels of severity [3,9,10,11]. Specifically, in MDD cognitive dysfunctions have been observed to persist despite remission of mood symptoms, suggesting an independence of cognitive symptoms compared to depressive symptoms [9,12,13]. However, individuals with SZ are much more likely to exhibit severe cognitive impairment than individuals with BD, and good cognitive functioning is more often observed in BD patients than in SZ patients [14]. Indeed, schizophrenia (SZ) represents one of the most severe and disabling mental disorders affecting 1% of the general population, which is characterized by a) positive symptoms, such as delusions, hallucinations, disorganization in thought or behaviors; b) negative symptoms, including flat affect, apathy, social withdrawal, poverty of thinking and speech; and c) cognitive symptoms, including poor executive functioning, difficulty maintaining attention and disturbance of working memory [2]. Moreover, different theories have been proposed to explain the neurobiological basis of cognitive impairment but none of them are completely comprehensive. Between these theories we found the neurodegeneration model correlated to dysregulation of the immune-inflammatory system, variations in the tryptophan catabolite (TRYCAT) pathway, dysregulation of glutamate signaling, reductions of neurotrophic elements such as brain-derived neurotrophic factor (BDNF) [15,16,17,18]. So, considering the lack of a clear biological basis for cognitive dysfunction in psychiatric disorders and the uncertain positive effect of antipsychotic, antidepressant/mood stabilizer drugs on cognitive impairment [9,19,20], there is a growing interest to find different neuroprotective and pro-cognitive agents for these disorders [21]. In this regard, the scientific community has recently driven attention towards the study of the ameliorating effects of nutrients and dietary supplementations on cognition by means of compounds that may be collectively named under the umbrella term of “functional nutrients”. 

### 1.2. Fatty Acids: The Biologic Plausibility of Docosahexaenoic Acid (DHA)

Among functional nutrients, *n*-3 polyunsaturated fatty acids (*n*-3 PUFAs) have been proposed as therapeutic supplements for a wide range of diseases as well as certain symptoms. Due to their pleiotropic properties, *n*-3 PUFAs are currently under investigation as a treatment for arteriosclerosis, cancer, diabetes, hypertension, arthritis, dementia, psychiatric disorders and some autoimmune diseases [22]. In the last two decades, there is accumulating evidence about the neuroprotective functions of LC-PUFAs, in particular docosahexaenoic acid (DHA), which is a *n*-3 LC-PUFAs together with eicosapentaenoic acid (EPA) and α-linolenic acid (ALA, the parental essential fatty acid of the *n*-3 series), as well as the *n*-6 LC-PUFA linoleic acid (LA), which is the essential parental *n*-6 fatty acid that is as an important constituent of neuronal membrane. ALA and LA are normally found in plant oils (e.g. walnut, edible seeds, clary sage seed oil, algal oil, flaxseed oil) while DHA and EPA are commonly found in marine oils (e.g. fish oils, eggs, krill oil). Marine algae and phytoplankton are primary sources of *n*-3 LC-PUFAs. LA and ALA from plants are the precursors (as previously indicated) of *n*-6 and *n*-3 LC-PUFAs, respectively. In human tissues, LA is converted mainly to arachidonic acid (AA), ALA to EPA and then less efficiently to DHA [23]. 

LC-PUFAs, particularly DHA and AA, are incorporated into cell membrane phospholipids and, apart from their structural role in these membranes, they can also modify the structure of the cell membrane and the function of membrane proteins acting as precursors of autocoid signaling molecules (e.g. docosanoids) and as potent activators of a number of gene transcription factors (e.g., peroxisome proliferator activated receptors). The most important role of *n*-3 LC-PUFAs is generally linked to the incorporation of DHA, in exceptionally high levels, in membranes of the central nervous system. Indeed, DHA is found in significant concentrations in the retinal and synaptic membranes due to its high fluidity [24]. 

Generally, the membrane structure of PUFAs—where the principal components are represented by LA, AA and DHA—seems to be more receptive to DHA coming from the diet than from the intake of LA and AA [25]. Moreover, data from animal models have shown that the increase of ALA from diet is directly proportional to membrane *n*-3/*n*-6 PUFA-ratios at LA/ALA intakes of <10. Instead, the dietary balance between ALA and LA influences slightly higher ALA intakes, and a similar biphasic response is also seen in diets that are rich in PUFAs [26]. These data demonstrate the high sensitivity of tissue membranes to variations in dietary PUFA supplies within the normal range, strongly preferring the inclusion of *n*-3 LC-PUFAs over LA and AA. Consequently, *n*-3 PUFAs seem to be the key element of membrane PUFAs composition and unsaturation. Indeed, the accumulation of DHA in membrane phospholipids cells of different tissues, e.g., erythrocytes (RBC), has been demonstrated to be related to diet, principally fish intake and breastfeeding during infancy, even though another small source of DHA is that formed endogenously by the desaturation and elongation of ALA. This biochemical process of conversion is limited by the delta-6 desaturase enzymatic step, which is normally less efficient, but the rate conversion has been shown to be influenced by the individual asset of the haplotypes, including single-nucleotide polymorphisms (SNPs), associated with PUFA metabolism. Indeed, in addition to diet, common polymorphisms in the fatty acid desaturase (FADS) gene cluster have very marked effects on LC-PUFA status [27].

For all these reasons, *n*-3 LC-PUFAs are considered to be one of the most important components of cell membranes, especially neurons, especially because they are implicated in the regulation of cross talk between cells and in metabolic processes of energy transformation. 

Additionally, LC-PUFAs are essential components for infant and child neurodevelopment because they are implicated in many neuronal processes (e.g. the regulation of membrane fluidity and gene expression) [27]. The accumulation of DHA in the brain starts in utero, especially during the second half of pregnancy, when DHA and AA accrue rapidly in the grey matter of the brain continuing up to two years of age and the high levels of DHA in the brain are maintained throughout life [28]. With respect to the functional effects of LC-PUFA supplementation in infancy, the most accepted developmental effect is an increased rate of visual acuity development [29]. Moreover, a deficiency or imbalance of *n*-3/*n*-6 LC-PUFAs has been associated with poorer child neurodevelopment in cognitive domains such as language (e.g. verbal fluency) and motor skills (e.g. gross and fine motor abilities) [30]. 

A growing body of evidence shows that diets with a lower intake of *n*-3 PUFAs, including EPA and DHA, are associated with the pathophysiology of different mental disorders [31]. On the contrary, cross-national epidemiological studies suggest that greater habitual dietary intake of fish/seafood, primary dietary sources of preformed EPA and DHA, are associated with reduced lifetime prevalence rates of affective disorders [32,33]. A meta-analysis of controlled trials suggested that short-term fish oil supplementation, especially with higher doses of EPA, is more effective than placebo to produce a beneficial overall effect in major depressive disorder patients [34]. Emerging translational evidence additionally suggests that DHA is required for normal brain development [27]; for this reason, a decrease in LC-PUFAs could have negative effects on cognitive function [35]. One possible explanation of the positive correlation between optimal DHA status and cognition may be due to the fact that during development, one of the most responsive cerebral areas to DHA supplementation is the frontal cortex [36]. The frontal lobes have a key role in executive and higher-order cognitive functions, including sustained attention, planning and problem solving [37], as well as for social, emotional and behavioural development [38]. Therefore, an optimal lipid DHA composition may not only be important during the development and maturation of the brain from pregnancy to childhood/adolescence but could also contribute to maintain cognitive efficiency during the entire lifespan and ameliorate the aging of the adult’s brain [39,40]. Indeed, in the last decades, there is accumulating evidence suggesting neuroprotective properties of *n*-3 LC-PUFAs, in particular DHA. Also, observational and clinical randomized trials have shown an association between high (“optimal”) levels of *n*-3 LC-PUFAs and a lower incidence of mental illnesses, such as depression [41,42]. Observational studies [43,44] showed a link between abnormalities in membrane fatty acid levels and the cognitive impairment in ultra-high-risk subjects for psychosis and in schizophrenic patients. Interestingly, a positive and significant correlation was found between the Brief Assessment of Cognition in Schizophrenia (BACS) composite score and blood EPA and DHA levels, while a negative and significant correlation was shown between a daily dose of antipsychotic medication, blood DHA levels and the BACS composite score. Moreover, neuroimaging studies also supported the positive link between *n*-3 PUFAs levels and mood functioning. Indeed, a structural Magnetic Resonance Imaging (MRI) study reported that individuals with higher dietary intake levels of *n*-3 PUFAs have greater grey matter volume in the anterior cingulate cortex, the right hippocampus, and the right amygdala, which are key areas involved in mood regulation and cognition [45]. 

## 2. Materials and Methods

A comprehensive search on PUBMED of all trails using DHA on cognitive functions in psychiatric disorders published up to November 2018 was performed.

Articles of potential interest were identified by using the following search terms: “omega-3“, “*n*-3 long-chain polyunsaturated fatty acids”, “*n*-3 LCPUFAs”, “DHA”, “docosahexaenoic acid”, combined with the following term: “psychiatric disorders”, “mental diseases”, “psychotic disorders”, “ psychosis”, “ultra-high risk psychosis”, “schizophrenia”, “bipolar disorder”, “major depressive disorder”, “affective disorder”, “depression”, “ personality disorder”, “ anxiety disorders”, “obsessive compulsive disorders”, “eating disorders”, “ADHD”, “autism” AND “ “cognitive functions”, “cognitive symptoms”, “cognition”. In this review, trials examining the efficacy of DHA on cognitive functions in psychiatric disorders were selected.

We considered only trials in which the authors used an exposure of DHA alone or combined with EPA as a unique treatment or as an adjunctive therapy to other drugs (e.g., antipsychotic, antidepressants, mood stabilizers and benzodiazepines), or other no pharmacological strategies such as psychotherapy and physical exercise compared to placebo or pharmacotherapy alone.

To limit the heterogeneity of this review and to reduce selection biases, we decided to exclude trials examining the efficacy of others *n*-3 LCPUFAs (e.g. EPA) in subjects with psychiatric diagnosis; trials analyzing serum levels of DHA as primary outcome; studies that did not explore the effects of DHA on cognitive functions or cognitive symptoms as primary outcome.

In addition, we excluded trials that employed a diet enriched in DHA as a supplementation.

Among the 1114 articles retrieved, 198 studies were identified and screened by reading the abstract, and, when necessary, the full text, in order to select those articles relevant for the analysis. A manual search of bibliographic cross-referencing complemented the search. Reference lists of relevant papers were also inspected to identify any additional trials.

Relevant articles were obtained and included in the review if (a) they reported an exposure to DHA; (b) included cognitive functions and cognitive symptoms as an outcome measure; and (c) enrolled human participants and reported a trial.

The process of identification and inclusion of trials is summarized in Figure 2. Finally, 8 trials were included for the review. All searches, trial identification, data abstraction, and tabulation were completed independently by three researchers. Discordances were discussed and resolved.

## 3. Results

### 3.1. DHA and Psychotic Disorders

Many data showed that cognitive symptoms are often an early marker and a potential predictor of outcome of illness in schizophrenia [46]. It is worth mentioning that approximately in 75%–85% of SZ patients showed cognitive deficits, especially in memory abilities [47]. Moreover, cognitive deficits have been found to be present already at the onset of illness and even in ultra-high risk (UHR) individuals [48,49] and in unaffected family members of patients [50,51], ultimately suggesting an important genetic contribution. While many theories regarding SZ pathogenesis have been proposed, the majority of the studies focused on neurotransmitter function, especially on abnormalities of dopamine activity. However, antipsychotic treatments based on dopamine hypothesis of schizophrenia have showed only a partial efficacy, and current drugs fail to alleviate symptoms in 30%–60% of patients. An alternative idea, postulated in 1970s, proposed a “phospholipid hypothesis” by which SZ could be considered a disorder of membrane phospholipid metabolism [52,53]. According to this theory, an increased activity of a phospholipase A2 (PLA2), which is a lipolytic enzyme, implicated a progressive and important loss of polyunsaturated fatty acids in neuronal membranes. This modification of membranes’ properties produced many biochemical abnormalities, such as reduced vasodilator responses to niacin and histamine and altered immunological functions. The final effect in the central nervous system is a membrane alteration that might be enough to produce the most serious consequences, since the brain needs co-ordinated sequential and parallel activities of millions of neurons. This hypothesis goes through abnormalities of *n*-6 fatty acids such as AA and of *n*-3 polyunsaturated fatty acids, such as DHA and EPA. In fact, many studies have shown a reduction of AA, DHA and EPA in the red blood cells (principal recognized markers for essential fatty acid status in the brain) of SZ patients [54,55]. Similarly, other studies also demonstrated that SZ patients had a depletion of PUFAs in plasma, thrombocytes and post-mortem tissue [56], as well as an elevation of membrane phospholipase A2 activity according to elevated membrane turnover of PUFAs [57]. Interestingly, the negative symptoms of SZ patients seem to be especially linked to severely abnormal levels of arachidonic acid and docosahexaenoic acid in red blood cell membrane phospholipids [58]. Overall, these studies reported several different alterations in neuronal membrane biochemistry, supporting the membrane hypothesis of schizophrenia in both chronic and first episode SZ patients [59]. In particular, SZ patients at the first episode often showed an increase in phospholipid membrane breakdown products within the brain [60], and this data is consistent with excitotoxic neural membrane breakdown and reduced levels of membranes phospholipid precursors, often observed in SZ [61]. We have then carried out a review of all randomized clinical trial studies (RCTs) with the final aim of providing a clearer picture of the impact of DHA in schizophrenic subjects and in ultra-high-risk (UHR) psychosis patients. Recent studies showed that early treatment of the prodromal period of psychotic disorders seems to be linked to more favourable outcomes [62]. Interestingly, among these treatments, *n*-3 PUFAs have been found to have potential beneficial impact due to their low incidence of adverse effects. To date, only four RCTs explored the role of *n*-3 PUFAs supplementation in UHR patients, but none of them investigated the cognitive effects of DHA as primary or secondary outcomes. Second, we explored clinical trials administering *n*-3 PUFAs versus placebo in schizophrenia patients for cognitive functioning but we found only two studies (one open label and one randomized controlled trial) using only EPA for supplementation [63,64] and none using DHA alone or combined with EPA, with contrasting results. In conclusion, taken all together these data suggested that no RCTs exploring the effect of DHA alone or with other *n*-3 PUFAs (EPA) or *n*-6 PUFAs have been conducted to investigate cognitive functioning in schizophrenic patients.

### 3.2. DHA and Mood Disorders

Bipolar disorder (BD) is a chronic mental illness characterized by recurrent episodes of mood alterations (depressed or maniacal features) [65], with several consequences on social and cognitive functioning [66]. Prevalence in general population ranges 2% to 3% considering the whole bipolar spectrum disorder [67]. Furthermore, it also affects patient’s families, with high health-related costs, medical morbidity and premature mortality [68]. A large number of BD patients (around 40% to 60%), present with cognitive impairments that impact everyday functioning, not only during maniacal or depressive episodes, but also during euthymia [69]. Patients with BD also have deficits in social skills, in fact the theory of the mind is clearly altered in maniacal phases but also altered in bipolar euthymic patients, and the same cognitive deficits could make the performance of social cognition worse [70]. Several factors related to the history of the bipolar disease seems to have a negative impact on memory performance and executive function, in particular, illness duration and severity of symptoms [71]. Moreover, a high number of mood episodes seems to be a predictor of poor performance in several cognitive domains. Scientific evidence underlined the presence of cognitive deficits in BD, also in remitted patients [72]. The intensity of the neuropsychological disturbances varies according to the severity of the disease; interestingly several findings indicate that people with BD II also show cognitive deficits of the same form, but slightly less severe than those observed in BD I. Cognitive deficits can be observed even before the onset of mental disease and executive functions seem to predict BD onset better than IQ [73]. Moreover, executive functions and other cognitive processes have been reported as being impaired in adults and youth with BD, even during remission [74]. Beyond executive functions, the most frequently compromised cognitive domains are attention, memory and verbal learning [75]. Therefore, longitudinal studies as well meta-analysis consistently reported the presence of cognitive deficits in BD, regardless of mood phase, ultimately suggesting that the assessment of cognition could contribute to a better understanding of the patient’s situation, pointing out possible vulnerabilities other than psychiatric symptoms. Since in BD patients’ clinically relevant deficits in cognition are often present, we could expect that a vicious loop to occur, given that executive function deficits may contribute to emotional dysregulation, which in turn would contribute to a decay in cognitive performance [76].

Many data, including functional and MRI (magnetic resonance imaging) and PET (positron emission tomography) studies, indicate the close link between emotion and cognition in BD; particularly neural activation deficits are shown in regions involved in responding to salient stimuli (amygdala, ventral striatum) and in the activation of regions involved in emotional regulation, such as the prefrontal cortical areas [77]. Diminished connectivity between areas involved in top-down control and areas involved in reactivity is also present. Furthermore, the activation of a frontoparietal circuit has been shown to correlate negatively with the activation of the amygdala during reappraisal of patients with BD type I. [78]. For these reasons, the study of the relation between cognition and emotion is a central starting point to understand the development of mood symptoms. Currently, there is growing interest in developing prevention strategies which have as primary outcome the management or prevention of neurocognitive impairment in BD [79]. In this perspective, recent literature has focused attention on the hypothetical positive effects of omega-3 polyunsaturated fatty acids in patients with affective disorders [80]. Despite the great interest of researchers for *n*-3 PUFAs and psychiatric diseases, just a few RCTs included neuropsychological evaluations or considered cognition as an outcome. Specifically, we identified only three RCT studies exploring the effect of *n*-3 PUFA supplementation on neuropsychological performance in mood disorder patients (Table 1). In all three studies, the authors did not find clear results; the score of neuropsychological assessments in patients treated with *n*-3 PUFA was not statistically different than patients taking placebo. In particular, two RCTs [81,82] explored the impact of *n*-3 PUFAs in a sample of depressed patients while the third trial [83] on a sample of patients with subthreshold depressive symptoms. In the latter, the authors have examined several neuropsychological domains using a standardized assessment in a cohort of 51 adults, taking EPA 1200 mg plus 800 mg DHA or placebo for a period of 12 weeks. They used the Trail Making Test Part A (TMT-A) and Part B (TMT-B), the Ray Auditory Verbal Learning Test (RAVLT) and the Delis Kaplan Executive Functioning System Stroop task in order to evaluate visuomotor speed, verbal learning and memory, and executive functions. Interestingly, the authors found no difference across all neuropsychological tests, in terms of performance, between the group taking *n*-3 PUFAs and placebo. Similarly, Rogers et al. [81] found no significant cognitive effects of supplementation with 630 mg EPA plus 850 mg DHA or placebo for 12 weeks in a larger sample of depressed patients (*n* = 190). Patients were tested with a neuropsychological assessment using the Visual probe task, the Simple reaction time, the Lexical decision, the Digit-symbol substitution, the Impulsivity score and the *N*-Back task. Authors targeted speed information processing, reasoning, impulsivity, and working memory. Finally, Antypa et al. [82] measured the cognitive performance of 71 depressed patients with neutral and emotional information processing tasks, a Facial Expression Recognition task and a Decision making (Gambling) task. Participants were treated with 1.74 g EPA plus 0.25 g DHA for a period of four weeks. Overall, the results were inconclusive. Indeed, only a small *n*-3 PUFA effect, though not statistically significant, was found on aspects of emotional decision-making and no significant effect was observed on memory, attention and cognitive reactivity. On the contrary, it is of note that the result of the Facial emotion recognition test in which the placebo group compared to the *n*-3 PUFA group was found to have a higher rate of accuracy in recognizing fear over time. This effect is not of trivial interpretation and deserves further investigation. In conclusion, no RCTs administering only DHA supplementation as add-on therapy in mood disorders have been published yet. However, we identified three RCTs on DHA combined with EPA supplementation in depressed patients, without, though, providing statistically significant results on the therapeutic effect of Omega 3 on cognitive performance. Therefore, it is necessary to provide clinical studies able to evaluate the response of mood and cognitive function together.

### 3.3. DHA and ADHD

ADHD is a clinical disorder characterized by symptoms of inattention, hyperactivity, and impulsivity [84]. ADHD is commonly diagnosed in childhood and has a high comorbidity with other disorders of behavior and mood, such as conduct disorder and developmental coordination disorder [85]. The worldwide prevalence of ADHD is estimated at 7.2% although there is significant variability between countries that may be attributed to differences in methodology between studies, such as diagnostic criteria [86]. The most common approaches to the treatment of ADHD are medication, mainly with stimulant medications methylphenidate (MPH) and amphetamine, and/or psychological or behavioral interventions [87]. However, not all children experience symptom reduction after pharmacological treatment with MPH. Therefore, although the responder rate to MPH is high (65–70%), supplementation with *n*-3 polyunsaturated fatty acids (*n*-3 PUFAs) might be an option for non- responders or treatment-resistant children [88]. Clinical and biochemical evidence suggests that functional deficiency of certain PUFAs could be related to ADHD. Children with ADHD have been shown to have significantly lower plasma and blood concentrations of PUFA and, in particular, lower levels of *n*-3 PUFA [89].

In this paragraph, we provide an update regarding the effects of *n*-3 PUFAs supplementation on symptoms of ADHD in children. We identified five RCT studies (Table 1).

Milte et al. investigated the effects of a selective supplementation with EPA or DHA versus the *n*-6 PUFA linoleic acid (LA) on behavior, cognition, and literacy in children 7 to 12 years old with ADHD symptoms [85].

A first analysis of results reported from the first four months of supplementation showed no significant treatment effects for literacy or parent-reported behavior with the Conners Parent Rating Scale (CPRS). Similarly, no significant treatment effects of DHA and EPA on the measurements of attention or inhibition were observed. 

The greatest benefit was observed in children who had comorbid learning difficulties: an increased erythrocyte DHA over four months was associated with improved word reading and oppositional behavior. In the 12-month study [90] the same authors found that changes in erythrocyte PUFA were associated with improvements in literacy (word reading and spelling), attention (Sky Search and Creature Counting), and behavior (subscales of parent-rated behavior through the CPRS). Particularly, increases in erythrocyte EPA, DHA, and total *n*-3 PUFA and decreases in erythrocyte *n*-6 and the *n*-6:*n*-3 ratio were associated with improvements.

The third work reported is a double-blind placebo-controlled intervention study carried out by Widenhorn-Müller et al. [91], which investigated the effect of *n*-3 fatty acid supplementation on behavior and cognition in children aged 6 to 12 years with ADHD. After 16 weeks of intervention, the authors showed that supplementation with EPA/DHA improved working memory function in the selected study population; however, no effects on other cognitive measures and parent- and teacher-rated behavior were pointed out.

In the same year, Dean et al. [92] published the results of the first trial specifically designed to assess the effects of fish oil supplementation on aggression in young people with impulsive aggressive behaviors. In particular, aggressive behavior includes verbal abuse, property destruction, and physical attacks and it can occur across a range of diagnoses, including disruptive behavior disorders, ADHD, autism, and intellectual disabilities. The primary aim of this study was to assess the efficacy of *n*-3 fatty acids (EPA+DHA) for at least six months in the treatment of impulsive aggression in children and adolescents aged 6–17 years with a diagnosis of a disruptive behavior disorder. The results showed that fish oil supplementation for six weeks had no beneficial effects on various ratings of aggressive behavior.

In conclusion, the most recent trial we report is by Crippa et al. [93] and, unlike the large majority of studies that explored the efficacy of PUFA supplementation in ADHD using mixed *n*-3 fatty acids, the aim of this research was to examine the effect of DHA supplementation as monotherapy in children aged 7 to 14 with ADHD. In general, no superiority of DHA supplement to placebo was observed on ADHD rating scale IV Parent Version–Investigator (ADHD-RS-IV), their a priori primary outcome. DHA supplementation showed a significant, nonetheless quite limited, effect on the hyperactivity–impulsivity scale and on total scores of the ADHD rating scale, with participants in both groups showing improvements over the six months. 

In conclusion, large high-quality RCTs are still required to further clarify the role of DHA on functional outcomes in this population and to draw conclusions.

## 4. Discussion 

The aim of this review was to provide a comprehensive overview of the effects of DHA on cognition in severe psychiatric disorders (Table 2). We found a growing number of RCTs testing the efficacy of *n*-3 PUFAs alone or as added therapy in the treatment of different psychiatric disorders. However, only a handful of studies investigated the efficacy of DHA alone on cognitive function. Indeed, the majority of RCTs on DHA focused on dementia or mild cognitive impairment [35,94]. Moreover, the small number of trials on cognitive impairment in psychiatric disorders show doubtful and/or contrasting results, probably due to the heterogeneity of the methods employed by the original studies, which often had small and inhomogeneous sample size, different selection criteria, different neuropsychological batteries, different carriers and dosages of DHA (i.e., alone, or a combination of DHA and EPA, or the addition of *n*-6 PUFAs) as well as various durations of supplementation. Only partial positive effects on specific tasks of neurocognitive batteries in the most important neuropsychiatric disorders have been shown, but even in this case the frequent lack of statistical adjustment for multiple comparisons makes most reported results doubtful.

In some cases, it is probable that the detection and treatment of n-3 PUFAs deficiency needs to be made early in the course of illness to exert a significant protection against the transition to psychosis and severity of symptoms [95,96,97]. For psychotic disorders, we did not find RCTs investigating the effect of DHA on cognition. Indeed, only two studies were conducted using EPA alone and the results were conflicting. While Reddy et al. [62] demonstrated a significative reduction in perseverative errors, which represents a key measure derived from the Wisconsin Card Sort Test (WCST), Fenton et al. (2001) [63] found no significant effects of EPA on ameliorating cognitive symptoms. However, these discordant results are due primarily to different designs of the study; one is an open label while the other is a clinical randomized trial. Other factors justifying the results are the different doses of EPA, different times of intervention, different sample sizes, different ages and durations of illness of the samples. 

Most studies considered here are limited by a small sample size and future research (particularly in the field of children with ADHD) should explore the consequences of *n*-3 PUFA supplementation in improving school performance, which is not necessarily a clinically diagnosed developmental disorder. 

For mood disorders, the reviewed studies have similar characteristics, all being quite well-designed and implemented. However, the appropriate dose of DHA to administer and the correct proportion between DHA and EPA [79] are still discussed. The heterogeneity of specific tests represents a great methodological limit as well. Longer periods of study should be considered, as well as a whole evaluation of clinical effects. [98,99,100,101,102,103,104,105,106]. The mechanism that binds mood, emotionality and cognitive abilities is very complex and the underlying neuroactive circuits have been only partially explained so far. The simultaneous lack of effect found on mood and cognition could suggest that these two aspects are firmly associated with each other, so neither aspect can improve regardless of the other. In our opinion, new research should take into consideration patients with different types of mood disorders. 

Future clinical trials should be focused on the role of *n*-3 PUFA supplementation in protecting neuropsychological functions, not only ameliorating psychiatric symptoms in patients treated with standard psychiatric therapies (antipsychotic, antidepressant), prolonging clinical observations at definitely longer terms.

## Figures and Tables

**Figure 1 nutrients-11-00769-f001:**
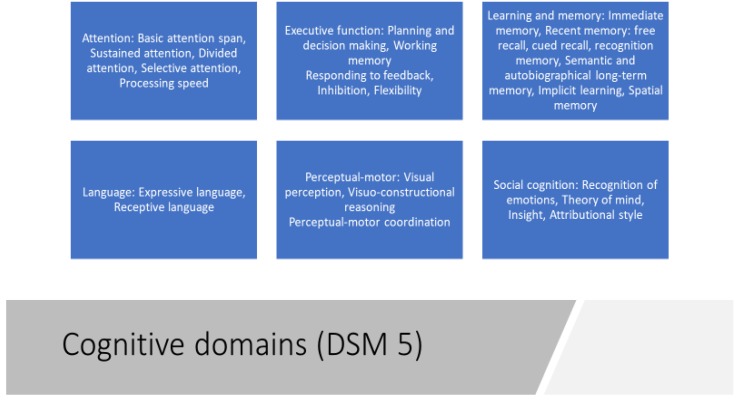
Cognitive domains according to Diagnostic and Statistical Manual of Mental Disorders (DSM-5).

**Figure 2 nutrients-11-00769-f002:**
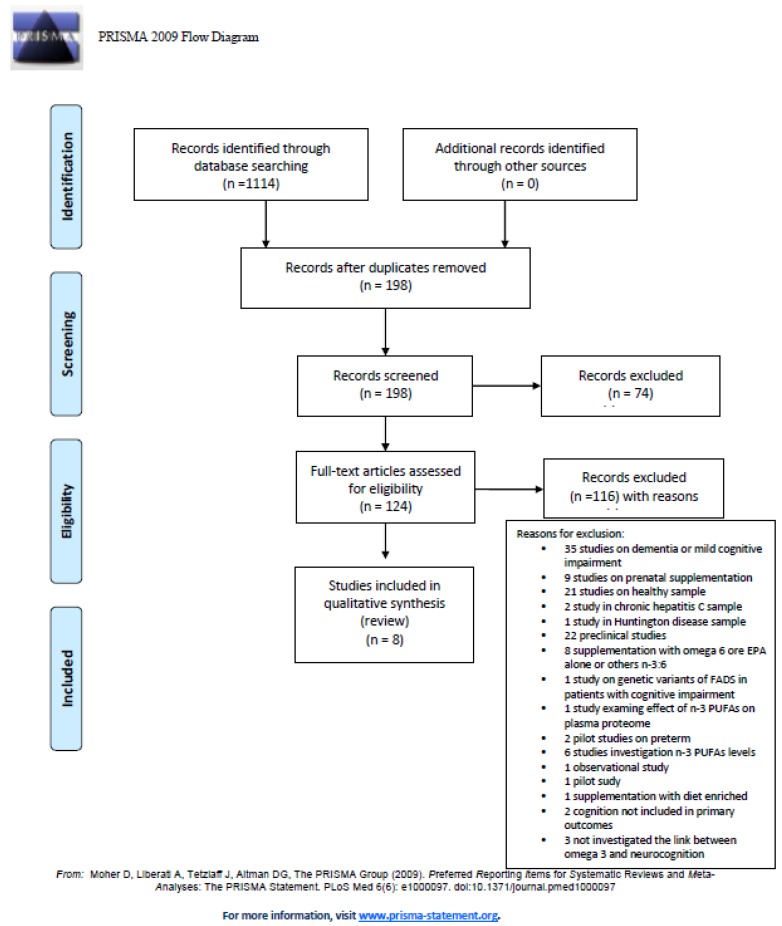
PRISMA diagram: docosahexaenoic acid (DHA) supplementation in psychiatric disorders.

**Table 1 nutrients-11-00769-t001:** Randomized controlled trials (RCTs) of *n*-3 polyunsaturated fatty acids (PUFAs) supplementation in mood and neurodevelopmental disorders.

Study	Diagnosis	N. Sample	*N*-PUFA	Duration	Neuropsychological Assessment	Major Finding
[80]	Mild to moderately depressed individuals	190	Three capsules/day contained a total of EPA 630 mg plus DHA 850 mg 870 mg olive oil mixed tocopherols 7.5 mg orange oil 12 mg Placebo treatment contained a total of Olive oil 2360 mg mixed tocopherols 7.5 mg orange oil 12 mg	12 weeks	Visual probe task. Simple reaction time. Lexical decision. Digit-symbol substitution. Impulsivity. *N*-Back.	No effects on cognitive function after *n*-3 LCPUFA supplementation.
[81]	Recovered depressed individuals	71	Three capsules/day contained a total of EPA 1.74 g DHA 0.25 g Placebo olive oil both of them lemon flavored	4 weeks	Neutral and emotional information processing tasks. Affective Go/No-Go task. Attentional Go/No-Go task. 15 Words test. Facial Expression Recognition task. Decision-making (Gambling) task.	No significant effects were observed on memory, attention, cognitive reactivity and depressive symptoms.
[82]	Patients “at risk” for major depression	51	Four capsules/day contained a total of EPA 1200 mg plus DHA 800 mg and one capsule of Sertraline Placebo 4 capsules/day of paraffin 1000 mg and one capsule of microcrystalline cellulose	12 weeks	Visuomotor speed (Trailmaking Test, Part A). Vebal learning and memory (The Rey Auditory Verbal Learning Test (RAVLT)) Executive functions (Trailmaking Test, Part B; Delis Kaplan Executive Functioning System (DKEFS)) Stroop task	No significant changes in any clinical or neuropsychological measures were found.
[88,89]	Children with attention-deficit/hyperactivity disorder	70 [89]; 90 [90]	4 × 500-mg capsules/day EPA-rich fish oil:EPA 1109 mg and DHA 108 mg 4 × 500-mg capsules/day DHA-rich fish oil: EPA 264 mg and DHA 1032 mg. Safflower oil (control):LA 1467 mg/day.	17 weeks [89], 52 weeks [90]	Literacy (Wechsler Individual Achievement Test III); parent-rated behavior (Conners Parent Rating Scale); different forms of attention (Test of Everyday Attention for Children)	No significant treatment effects for literacy; parent-reported behavior; measurements of attention or inhibition.
[90]	Children with attention-deficit/hyperactivity disorder	95	720 mg *n*-3 fatty acids (600 mg EPA, 120 mg DHA). Placebo (olive-oil)	16 weeks	Behavior (FBB ADHS parent-rated and teacher-rated questionnaires, DIS- YPS-II; German version of the Child Behavior Check list; German version of the Teacher's Report Form)	Significant improvement of working memory function (Index Score) when comparing the EPA/DHA group with the placebo-taking group.
[91]	Young people with impulsive aggressive behaviors.	21	400 mg EPA and 2000 mg DHA Placebo (low polyphenol olive oil and 10 mg standard fish oil)	6 weeks	Aggressive behavior (The Children’s Aggression Scale-Parent Version, CAS-parent; The Modified Overt Aggression Scale, MOAS)	No effect of treatment on primary ratings of aggressive behavior
[92]	Children with attention-deficit/hyperactivity disorder	50	500 mg algal DHA Placebo (500 mg wheat germ oil)	26 weeks	Behavior (ADHD rating scale IV Parent Version–Investigator)	No beneficial effect on the symptoms of ADHD

**Table 2 nutrients-11-00769-t002:** Effects of *n*-3 PUFAs supplementation in mood and neurodevelopmental disorders: summary.

Diagnosis	Positive Results	Negative Results	Positive Results without Statistical Significance
Mild to moderately depressed individuals		[80]	Cognitive performance in the impulsivity task might be improved
Recovered depressed individuals		[81]	
Patients “at risk” for major depression		[82]	
Children with ADHD		[89]	
Children with ADHD		[85]	
Children with ADHD	[90]		
Young people with impulsive aggressive behaviors		[91]	
Children with ADHD		[92]

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
