# Peer review of "The Role of Docosahexaenoic Acid (DHA) on Cognitive Functions in Psychiatric Disorders"

_nutrients, 2019, doi:10.3390/nu11040769_

Round 1
Reviewer 1 Report
The present paper reviewed the literature on the relationship between DHA and cognitive functions in mental disorders. Studies included in the review were comprehensive. However, the purpose of the present paper was not well justified; much information presented in the paper might not be necessary and irrelevant to the focus; and the organization of the materials needs to be improved.
First of all, the rationale of reviewing this literature was not clear. On p.6, the authors stated "... the growing number of studies on the effects of .... reflects the growing interest in this area of research". This would serve as one reason to conduct the present study, but this was not enough to justify it. Readers would expect more on the importance of DHA in clinical research on psychiatric disorders. However, the authors only introduced neurochemical mechanisms, without explicitly discussing the importance of DHA.
Second, the authors spent too much time on basic concepts and physiological/neurochemical mechanisms. For example, the whole section of 1.1 introduced human cognition; much of 2.1-2.3 covered cognitive functioning in psychiatric disorders. This information is important to prepare readers for later sections. However, since the audience would be likely to already have knowledge background in this area, the introduction of the background knowledge should be brief. A considerable amount of the information might be unnecessary. On the other hand, some important data (e.g., epidemiological research) were missing.
Last, the organization of materials did not help readers obtain useful information from the paper. Specifically, the paper started with introducing human cognition. This was less relevant, and made the paper loss the focus. It would be better to start with cognitive functions in psychiatric disorders (i.e., section 1.2). Also, the same information was presented repeatedly in Introduction and Results section (e.g., cognition in psychiatric disorders). Another problem is that Materials and Methods section was placed after Results. Given this paper is a qualitative review, there is no need to structure the paper as the way it is. If the structured paper format is used, it would be clearer that Results section goes after the Methods section. Discussion section is also needed. Additionally, each paragraph was a little too long, readers would not follow the stream of thoughts easily.
Author Response
Reviewer#1:
General: The present paper reviewed the literature on the relationship between DHA and cognitive functions in mental disorders. Studies included in the review were comprehensive. However, the purpose of the present paper was not well justified; much information presented in the paper might not be necessary and irrelevant to the focus; and the organization of the materials needs to be improved.
RESPONSE: We thank the reviewer for summarizing our study. All the reviewer's comments have been taken into consideration and, we believe, have greatly ameliorated the quality of the revised version of the manuscript. We thank the referee for sharing her/his thoughts and having spent her/his precious time on reviewing the manuscript.
Details:
First of all, the rationale of reviewing this literature was not clear. On p.6, the authors stated "... the growing number of studies on the effects of .... reflects the growing interest in this area of research". This would serve as one reason to conduct the present study, but this was not enough to justify it. Readers would expect more on the importance of DHA in clinical research on psychiatric disorders. However, the authors only introduced neurochemical mechanisms, without explicitly discussing the importance of DHA.
RESPONSE: We thank the reviewer for highlighting this aspect. We clarified the importance of DHA and its possible role in mental health in the paragraph titled “1.3 Fatty acids: the biologic plausibility of docosahexaenoic acid (DHA)” where we underlined the importance of this fatty acid in the structure of cell membrane (specifically in the brain), synaptic plasticity, child neurodevelopment (See references from 22 to 33). Moreover, in the last part of the paragraph we suggested the correlation between DHA and cognition and mental illness (see references from 34 to 41). We agree with the reviewer about the explication of the reason for our work, therefore we modified the last sentence of the text as follows:
“The growing number of studies on the association of n-3 LC-PUFAs on cognitive function and mental health, especially psychiatric illness, published in the last decades, reflects the growing interest in this area of research. Given the global market for n-3 PUFAs products it is of public importance that there is a more conclusive picture as to whether their supplementation improves cognitive performance. Therefore, we conducted a review of randomized placebo-controlled trials (RCTs) which examined the role of n- 3 PUFAs, in particular DHA, in maintaining or in treating cognitive functions in different mental disorders.”
Second, the authors spent too much time on basic concepts and physiological/neurochemical mechanisms. For example, the whole section of 1.1 introduced human cognition; much of 2.1-2.3 covered cognitive functioning in psychiatric disorders. This information is important to prepare readers for later sections. However, since the audience would be likely to already have knowledge background in this area, the introduction of the background knowledge should be brief. A considerable amount of the information might be unnecessary. On the other hand, some important data (e.g., epidemiological research) were missing.
RESPONSE: Thank you very much for pointing this out. We agree with the reviewer’s comment and the section 1.1 and 2.1 have been shortened, making the introduction briefer. Then we included a lot of specific sentence about epidemiological studies conducted on PUFAs in mental illness as follows:
“A growing body of evidence showed that a diet with a lower intake of n-3 PUFAs, including EPA and DHA, is associated with the pathophysiology of different mental disorder [31]. On the contrary cross-national epidemiological studies suggest that greater habitual dietary intake of fish/seafood, primary dietary sources of preformed EPA and DHA, are associated with reduced lifetime prevalence rates of affective disorders [32-33]. A meta-analysis of controlled trials suggested that short-term fish oil supplementation, especially higher doses of EPA is more effective than placebo to produce a beneficial overall effect in major depressive disorder patients [34]. Emerging translational evidence additionally suggests that DHA is required for normal brain development [27]; for this reason, a decrease of LC-PUFAs could have negative effects on cognitive function [35]. One possible explanation of the positive correlation between optimal DHA status and cognition may be due to the fact that during development one of the most responsive cerebral area to DHA supplementation is represented by the frontal cortex [36]. The frontal lobes have a key role for executive and higher-order cognitive functions, including sustained attention, planning and problem solving [37], as well as for social, emotional and behavioural development [38]. Therefore, an optimal lipid DHA composition may not only be important during the development and maturation of the brain from pregnancy to childhood\adolescence but could also contribute to maintain a cognitive efficiency during the entire life and ameliorate the aging of the adult’s brain [39,40]. Indeed, in the last decades, there are accumulating evidence suggesting neuroprotective properties of n-3 LC-PUFAs, in particular of DHA. Also, observational and clinical randomized trials have shown an association between high (“optimal”) levels of n-3 LC-PUFAs and a lower incidence of mental illnesses, such as depression [41, 42]. Observational studies [43, 44] showed a link between abnormalities in membrane fatty acid levels and the cognitive impairment in ultra-high risk subjects for psychosis and in schizophrenic patients. Interestingly, a positive and significant correlation was found between the Brief Assessment of Cognition in Schizophrenia (BACS) composite score and blood EPA and DHA levels, while a negative and significant correlation was showed between a daily dose of antipsychotic medication and blood DHA level and the BACS composite score. Moreover, neuroimaging studies also supported the positive link between n-3 PUFAs levels and mood functioning. Indeed, a structural Magnetic Resonance Imaging (MRI) study reported that individuals with higher dietary intake levels of n-3 PUFAs have greater grey matter volume in the anterior cingulate cortex, the right hippocampus, and the right amygdala, which are key areas involved in mood regulation and cognition [45]. The growing number of studies on the association of n-3 LC-PUFAs on cognitive function and mental health, especially psychiatric illness, published in the last decades, reflects the growing interest in this area of research. Given the global market for n-3 PUFAs products it is of public importance that there is a more conclusive picture as to whether their supplementation improves cognitive performance. Therefore, we conducted a review of randomized placebo-controlled trials (RCTs) which examined the role of n- 3 PUFAs, in particular DHA, in maintaining or in treating cognitive functions in different mental disorders.”
Last, the organization of materials did not help readers obtain useful information from the paper. Specifically, the paper started with introducing human cognition. This was less relevant, and made the paper loss the focus. It would be better to start with cognitive functions in psychiatric disorders (i.e., section 1.2). Also, the same information was presented repeatedly in Introduction and Results section (e.g., cognition in psychiatric disorders). Another problem is that Materials and Methods section was placed after Results. Given this paper is a qualitative review, there is no need to structure the paper as the way it is. If the structured paper format is used, it would be clearer that Results section goes after the Methods section. Discussion section is also needed. Additionally, each paragraph was a little too long, readers would not follow the stream of thoughts easily.
RESPONSE: Thank you very much for pointing this out. In accordance to reviewer's input, we modified and reduced the introduction and we moved the “Materials and Methods” section after the Introduction.
Concerning the “Discussion”, we argued all the results inside each paragraph related to specific disease and we summarized our conclusions and proposed future direction in the last paragraph titled “CONCLUSIONS AND FUTURE DIRECTION”, in particular these sentences clarify our suggestions to better investigate this topic:
“The simultaneous lack of effect found on mood and cognition could suggest that these two aspects are firmly associated with each other, so neither aspect can improve regardless of the other. In our opinion the new research scenario should take into consideration patients with different types of mood disorders. Future clinical trials should be focused on the role of n-3 PUFAs supplementation in protecting neuropsychological functions and not only ameliorating psychiatric symptoms in patients treated with standard psychiatric therapies (antipsychotic, antidepressant) prolonging clinical observations at definitely longer term”
Reviewer 2 Report
The initial sections describing cognitive domains and fatty acid biochemistry are extensive, but probably too extensive for this review. These sections account for five of the fourteen pages of text.
I also have some confusion of the number of studies reviewed. The method section indicates that only 8 papers have been reviewed in this manuscript, but the text suggests somewhat more. There are seven studies referring to mood and developmental disorders and more than one relating to schizophrenia.
The English language in understandable but needs editing for syntax and grammar.
Author Response
Reviewer#2:
General: The initial sections describing cognitive domains and fatty acid biochemistry are extensive, but probably too extensive for this review. These sections account for five of the fourteen pages of text.
RESPONSE: All the reviewer's comments have been taken into consideration and, we believe, have greatly ameliorated the quality of the revised version of the manuscript. We thank the referee for sharing her/his thoughts and having spent her/his precious time on reviewing the manuscript.
I also have some confusion of the number of studies reviewed. The method section indicates that only 8 papers have been reviewed in this manuscript, but the text suggests somewhat more. There are seven studies referring to mood and developmental disorders and more than one relating to schizophrenia.
RESPONSE: Thank you very much for pointing this out. The correct number is 8 RCTs totally: 5 for neurodevelopment and 3 for mood disorders. We agree with the reviewer’s comment about the number of studies discussed in the text that in some paragraph differs from the number described in materials and methods, specifically for schizophrenia. In this section of results, we have discussed different observational studies on levels of DHA and EPA in schizophrenia and two RCTs using EPA supplementation and we have declared that about the supplementation of DHA alone or in combination with EPA none RCT studies was conducted in schizophrenic patients. Actually, this could be misleading for the audience, so we modified it, according to reviewer’s comment. Specifically, the paragraph titled “2.1 DHA AND PSYCHOTIC DISORDERS” has been changed as follows:
“Schizophrenia (SZ) is one of the most severe and disabling mental disorder affecting 1% of the general population, which is characterized by a) positive symptoms, such as delusions, hallucinations, disorganization in thought or behaviours, b) negative symptoms, including flat affect, apathy, social withdrawal, poverty of thinking and speech and c) cognitive symptoms, including poor executive functioning, difficult on maintain attention and disturbance of working memory [2]. Notably, cognitive symptoms are often an early marker and a potential predictor of outcome of illness [46]. It is worth mentioning that approximately in 75-85% of SZ patients showed cognitive deficits, especially in memory abilities [47]. Moreover, cognitive deficits have been found to be present already at the onset of illness and even in ultra-high risk (UHR) individuals [48, 49] and in unaffected family members of patients [50, 51], ultimately suggesting an important genetic contribution. While many theories regarding SZ pathogenesis have been proposed, the majority of the studies focused on neurotransmitter function, especially on abnormalities of dopamine activity. However, antipsychotic treatments based on dopamine hypothesis of schizophrenia have showed only a partial efficacy, and current drugs fail to alleviate symptoms in 30-60% of patients. An alternative idea, postulated in 1970s, proposed a “phospholipid hypothesis” by which SZ could be considered a disorder of membrane phospholipid metabolism [52-53]. According to this theory, an increased activity of a phospholipase A2 (PLA2), which is a lipolytic enzyme, implicated a progressive and important loss of polyunsaturated fatty acids in neuronal membranes. This modification of membranes’ properties produced many biochemical abnormalities, such as reduced vasodilator responses to niacin and histamine and altered immunological functions. The final effect in nervous central system is a membrane alteration that might be enough to produce most serious consequences, since the brain needs co-ordinated sequential and parallel activities of millions of neurons. This hypothesis goes through abnormalities of n-6 fatty acids such as AA and of n-3 polyunsaturated fatty acids, such as DHA and EPA. In fact, many studies have showed a reduction of AA, DHA and EPA in red blood cell (principal recognized markers for essential fatty acids status in the brain) of SZ patients [54, 55]. Similarly, other studies also demonstrated that SZ patients had a depletion of PUFAs in plasma, thrombocytes and post-mortem tissue [56] as well as an elevation of membrane phospholipase A2 activity according to elevated membrane turnover of PUFAs [57]. Interestingly, negative symptoms of SZ patients seem to be especially linked to severely abnormal levels of arachidonic acid and docosahexaenoic acid in red blood cell membrane phospholipids [58]. Overall these studies reported several different alterations in neuronal membrane biochemistry, supporting the membrane hypothesis of schizophrenia in both chronic and first episode SZ patients [59]. In particular, SZ patients at the first episode often showed an increased phospholipid membrane breakdown products within the brain [60], and this data is consistent with excitotoxic neural membrane breakdown and reduced levels of membranes phospholipid precursors, often observed in SZ [61]. We have then carried out a review of all randomized clinical trial studies (RCTs) with the final aim of providing a clearer picture of the impact of DHA in schizophrenic subjects and in ultra-high risk (UHR) psychosis patients. Recent studies showed that early treatment of the prodromal period of psychotic disorders seems to be linked to more favourable outcomes [62]. Interestingly, among these treatments, n-3 PUFAs have been found to have potential beneficial impact due to their low incidence of adverse effects. To date, only four RCTs explored the role of n-3 PUFAs supplementation in UHR patients, but none of them investigated the cognitive effects of DHA as primary or secondary outcomes. Second, we explored clinical trials administering n-3 PUFAs vs placebo in schizophrenia patients for cognitive functioning but we found only two studies (one open label and one randomized controlled trial) using only EPA for supplementation [63, 64] and none using DHA alone or combined with EPA. with contrasting results. In conclusion, taken all together these data suggested that no RCTs exploring the effect of DHA alone or with other n-3 PUFAs (EPA) or n-6 PUFAs has been conducted to investigate cognitive functioning in schizophrenic patients.
The English language in understandable but needs editing for syntax and grammar.
RESPONSE: We thank the referee for this input. We have carefully reread the whole text and we have edited the English language and style.
Round 2
Reviewer 1 Report
The authors addressed the issues raised in the comments and made substantial changes in the manuscript. The materials are now well organized in the revision. The paper contributes to the literature, and is potentially important for future studies of DHA and cognition in psychopathology.
My concerns about the first version have been addressed in the authors' responses. Here, I list my comments on the revision:
In the Introduction, it would be better to give an overview of the study in a short paragraph or few sentences.
In Results section, the information should match Methods. Still, some paragraphs in Results should be moved to an earlier section. For example, the sentence "Schizophrenia (SZ) is one of the most severe and disabling ..." (3.1, on p.9) is not a result form the search and synthesis of the literature on this specific topic, which should be stated in Introduction.
Some psychological disorders (e.g., PTSD, OCD) are mentioned in the Introduction, but not discussed in later sections.
The title of the subsection 3.3 is DHA and Neurodevelopmental Disorders. However, only ADHD, not other neurodevelopmental disorders, was covered. It's more precise to use "DHA and ADHD" as the title of the subsection.
There are still some typos and errors, e.g., "obsessive-compulsive disorder (DOC)" (p.5), it should be "OCD" ; "Information process starts with ..." (p.3), it should be "Information processing ..."; "Schizophrenia (SZ) is one of the most severe and disabling mental disorder ..." (p.9) should be "... mental disorders ..."; etc.
Author Response
The authors addressed the issues raised in the comments and made substantial changes in the manuscript. The materials are now well organized in the revision. The paper contributes to the literature, and is potentially important for future studies of DHA and cognition in psychopathology.
My concerns about the first version have been addressed in the authors' responses. Here, I list my comments on the revision:
In the Introduction, it would be better to give an overview of the study in a short paragraph or few sentences.
RESPONSE: Thank you for this input. In accordance to reviewer's input, this paragraph has now been modified, page 1, as follows:
“The growing number of studies on the association of n-3 LC-PUFAs on cognitive function and mental health, especially psychiatric illness, published in the last decades, reflects the growing interest in this area of research. Given the global market for n-3 PUFAs products it is of public importance that there is a more conclusive picture as to whether their supplementation improves cognitive performance. Therefore, we conducted a review of randomized placebo-controlled trials (RCTs) which examined the role of n- 3 PUFAs, in particular DHA, in maintaining or in treating cognitive functions in different mental disorders.”
In Results section, the information should match Methods. Still, some paragraphs in Results should be moved to an earlier section. For example, the sentence "Schizophrenia (SZ) is one of the most severe and disabling ..." (3.1, on p.9) is not a result form the search and synthesis of the literature on this specific topic, which should be stated in Introduction.
RESPONSE: Thank you very much for pointing this out. We moved the sentence, that you suggested, from result to introduction, page 3, now you can read as follows:
“ Indeed schizophrenia (SZ) represents one of the most severe and disabling mental disorders affecting 1% of the general population, which is characterized by a) positive symptoms, such as delusions, hallucinations, disorganization in thought or behaviours, b) negative symptoms, including flat affect, apathy, social withdrawal, poverty of thinking and speech and c) cognitive symptoms, including poor executive functioning, difficult on maintain attention and disturbance of working memory [2].”
Some psychological disorders (e.g., PTSD, OCD) are mentioned in the Introduction, but not discussed in later sections.
RESPONSE: Thank you very much for pointing this out, even if the raison is that to this day no study has been conducted on the effect of DHA on cognition in these psychological disorders, but only on the effect of DHA on psychopathological symptoms of PTSD or OCD so we could not discuss them in the later sections.
The title of the subsection 3.3 is DHA and Neurodevelopmental Disorders. However, only ADHD, not other neurodevelopmental disorders, was covered. It's more precise to use "DHA and ADHD" as the title of the subsection.
RESPONSE: Thank you for this input. In accordance to reviewer's input, the title of the subsection 3.3 has now been modified, page 10, as follows:
“3.3 DHA AND ADHD”
There are still some typos and errors, e.g., "obsessive-compulsive disorder (DOC)" (p.5), it should be "OCD” ; "Information process starts with ..." (p.3), it should be "Information processing ..."; "Schizophrenia (SZ) is one of the most severe and disabling mental disorder ..." (p.9) should be "... mental disorders ..."; etc.
RESPONSE: Thank you very much for pointing this out. We corrected all the typos and errors that you suggested.
Reviewer 2 Report
This revised manuscript is a significant improvement over the original. There is still need for minor editing, but the meaning is clear.
Author Response
This revised manuscript is a significant improvement over the original. There is still need for minor editing, but the meaning is clear.
RESPONSE: We thank the referee for appreciating our revised manuscript. We have now modified some sentences and grammatical errors to make it ready for publication.